# The association of muscle weakness with functional disability in older patients with Diabetes mellitus: Measured by three different grip strength thresholds

Sibel Çavdar ¤a*, Fatma Ozge Kayhan Kocak ¤b, Sumru Savas

Division of Geriatrics, Department of Internal Medicine, Ege University Hospital, Izmir, Turkiye

¤a Current address: Division of Geriatrics, Department of Internal Medicine, Izmir City Hospital, Izmir, Turkiye
¤b Current address: Division of Geriatrics, Department of Internal Medicine, University of Health Sciences Tepecik Education and Research Hospital, Izmir, Turkiye
* sibelilgun@gmail.com

**Citation:** Çavdar S, Kocak FOK, Savas S (2025) The association of muscle weakness with functional disability in older patients with Diabetes mellitus: Measured by three different grip strength thresholds. PLoS ONE 20(1): e0317250. https://doi.org/10.1371/journal.pone.0317250

**Data Availability Statement:** There are ethical restrictions on publicly sharing a de-identified minimal data set due to sensitive patient information. Data is available from the secretary of

## Abstract

The association of muscle weakness with poor outcomes is well defined in general older population, but there is insufficient data on the association of muscle weakness with functionality in older patients with diabetes mellitus (DM). We aimed to investigate the predictivity of muscle weakness defined as low grip strength thresholds determined by EWGSOP2, and two regional thresholds in older patients with DM for functional disability. Activities of Daily Living (ADL), Instrumental ADL (IADL), grip strength, comorbidities, anthropometric and biochemical data from outpatient clinic medical records were screened retrospectively. Low grip strength was determined by EWGSOP2, and two regional thresholds. Receiver operating characteristic (ROC) analysis, sensitivity and negative predictive values were conducted to identify the discrimination power of three different grip strength thresholds for functional disability in patients with DM. A total of 197 patients with DM and 215 controls were included. In ROC analyses, regional thresholds were with higher sensitivity and negative predictive values for functional disability in both groups. For patients with DM, regional normative thresholds predicted functional disability both for ADLs and IADLs whereas for patients without DM normative thresholds predicted ADL, and calculated thresholds predicted IADL disability. Regional normative thresholds predicted both ADL and IADL functional disability in older patients with DM.

## 1. Introduction

Diabetes mellitus (DM) is a widespread health problem for the aging population, and number of older adults with DM is projected to increase over the next few decades [1, 2]. Globally, metabolic risks and behavioral factors such as low physical activity that has a bidirectional relationship with muscle strength contributed the most attributable death and disability-adjusted life-years of diabetes [3]. Diabetes has negative impact on muscle strength [1], and hyperglycemia

the Ege University Ethics Committee via email (current affiliationegetaek@gmail.com) for researchers who meet the criteria for access to confidential data.

**Funding:** The author(s) received no specific funding for this work.

**Competing interests:** The authors have declared that no competing interests exist.

is suggested to be relevant to muscle weakness in patients with DM [4]. Older individuals with muscle weakness have a higher risk of poor outcomes such as longer hospital stay, functional disability and mortality [5], whereas functional disability is also associated with poor prognosis and increasing costs [6]. Besides, though functionality is substantial for older individuals, evaluating functionality is particularly essential in older patients with DM as it takes place in most of the treatment and follow-up decision processes as well as adherence to walking and exercise recommendations. Diabetes management in underserved individuals, including older adults, is largely based on self-management, and it refers to the activities and behaviors an individual undertakes to control and treat their condition [7]. Herein, measurement of grip strength might be an objective and easy method to perform for functional status. In addition, muscle weakness may be an important predictor of falls and impairment in activities of daily living in the older patients with DM. Higher muscle strength is associated with a lower risk for the impairment of Activities of Daily Living (ADL) and Instrumental ADL (IADL) [8, 9]. Low grip strength is associated with worsening of functional disability such as ADLs, and IADLs in older population [10, 11]. Besides, DM is an important risk factor and predictor for functional disability in older patients [6, 9, 12]. However, the association of grip strength with functional status in older patients with DM is poorly investigated [13–15]. Muscle strength measurement by grip strength is a simple and inexpensive method [5, 16]. However, various cut-off points and methods are used to evaluate low grip strength values which makes it difficult to compare the results [17, 18]. Those thresholds depend on measurement techniques, reference studies, and populations [16]. The European Working Group on Sarcopenia in Older People (EWG-SOP) indicated a sarcopenia definition in 2010 and than in 2018 the group updated the definition. The revised EWGSOP (EWGSOP2) consensus for the definition and diagnosis of sarcopenia has suggested cut-off points to provide harmonization among studies [16]. In this consensus paper, the use of regional normative populations is also recommended when available as measurements such as gait speed and strength depend upon stature [16]. Several cut-off points derived by different methods for low grip strength were suggested for the Turkish population [19–21] such as the thresholds determined by the use of regional normative populations [20], and those thresholds driven from the gait speed measurements [19]. Besides, the best low grip strength thresholds to predict functional disability comparing the EWGSOP2 consensus thresholds with regional data determined by different methods have not been studied in general older population and also in older patients with DM. Therefore, we aimed to investigate the predictivity of muscle weakness for functional disability by three different low grip strength thresholds in older patients with DM in comparison with each other and also in comparison with older patients without DM. Pointing out those muscle weakness thresholds will also be practical for clinicians in terms of screening functional disability with an objective measurement. Thus, the patients with or without the need of closer follow-up for functional disability might be defined by the identified grip strength thresholds.

## 2. Materials and methods

We screened patients from hospital records between October 2015 and 2019 in geriatric outpatient clinic. Data was scanned from the system between 10.01.2023 and 28.03.2023. Authors did not have access to information that could identify individual participants during or after data collection. In this retrospective study, both diabetic and non-diabetic patients without acute conditions as well as malignancy were included. Patients with missing ADLs or IADLs, and grip strength values, having sensory (vision, hearing), and cognitive impairments that might affect communication were excluded. Patients with severe pain, arthritis, or deformity in the hand joints, which could compromise handgrip strength, were excluded from the study

as handgrip strength measurements were not conducted in such cases. In terms of patients with DM, individuals with neuropathy and retinopathy to influence measurement of GS, were also excluded. Besides patients diagnosed with DM for at least 1 year were included. The exclusion and inclusion criteria for DM, and non-DM groups, and number of the patients are shown in **Fig 1**.

Socio-demographic data (age and gender), fasting blood glucose, creatinine, HbA1c, body mass index (BMI), number of medications, comorbidities, grip strength values, and functional status were noted in all patients. All patients' data were retrieved from hospital records retrospectively.

Muscle strength was assessed by grip strength measured by Takei T.K.K. 5401 digital dynamometer (Takei Scientific Instruments Co. Ltd, Tokyo, Japan) with a validated protocol by the National Health and Nutrition Examination Survey [22]. The maximum value of three grip strength measurements was selected. Low grip strength was determined by three distinct methods; i) EWGSOP2 consensus thresholds [16] (for males <27 kg, and for females <16 kg), ii) regional calculated thresholds derived from gait speed by Receiver operating characteristic (ROC) analysis [19] (for males <32 kg, and females <22 kg), and iii) normative regional thresholds [20] (for males <35 kg, and females <20 kg).

Functional status was measured by KATZ ADLs [23], and Lawton-Brody IADLs [24]. In the hospital records total scores for ADLs and IADLs scales were 6 and 23 points, respectively [25, 26]. The IADL forms in the patient files were re-examined and re-scoring was performed based on the current valid IADL score of 8 points [27]. Patients were evaluated as "having disability" by ADLs and IADLs scales, if scores were <5, and <8, respectively.

The calculations of the sample power were performed by the Post Hoc Compute Achieved Power analysis using the G* Power software version 3.0.10 [28]. With a sample size of 197

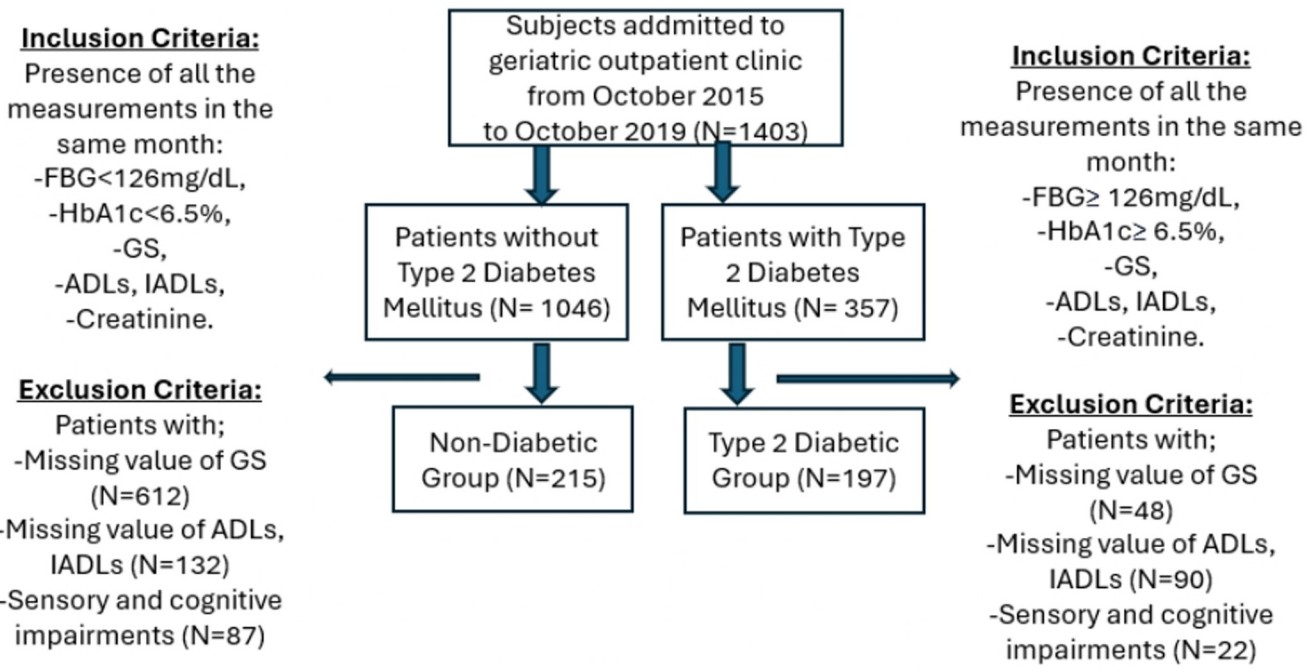

**Fig 1. Flow chart of patients enrolled in the study.** The exclusion and inclusion criteria for DM, and non-DM groups, and number of the patients are shown. FBG; Fasting blood glucose, HbA1c; Hemoglobin A1c, GS; Grip strength, ADLs; Activities of Daily Living, IADLs; Instrumental Activities of Daily Living.

patients with DM, an effect size of 0.5, and a margin of error of 0.05, the calculated representation power was 0.99.

Normality was assessed by Shapiro-Wilk test. Normally distributed quantitative variables were expressed by mean±standard deviation, and those without normal distribution were expressed by median and minimum-maximum values. Chi-Square ($x^2$) test and Fisher's exact test were used in analysis of categoric variables, and those were expressed as frequency and percentages. The *t*-test and Mann-Whitney U test were used in analysis of quantitative variables where available.

Multivariate analysis included parameters that were significant in presence of disability such as hyperlipidemia, cardiovascular disease, polypharmacy and BMI. Multiple linear regression, the "enter" method, was used to define descriptive variables on disability, and to calculate the adjusted odds ratios (ORs) and 95% confidence interval (CI).

To evaluate diagnostic performance of three identified grip strength cut-offs, the following parameters were calculated: sensitivity, specificity, positive predictive value, negative predictive value, as well as the area under the ROC curve (AUC). We used the 2×2 cross-tabulation for calculating specificity, sensitivity, positive predictive value, and negative predictive value at the diagnostic accuracy of GS with different thresholds for functional disability [29]. Sensitivity and specificity were classified as good (>80%), fair (50–80%), or poor (<50%) [30]. The AUC is a measure of the overall diagnostic accuracy of a test. An AUC>0.8 indicates good, 0.6–0.8 fair, and <0.6 indicates poor diagnostics accuracy [31]. We adopted criteria focused on sensitivity and negative predictive value which minimizes false negative outcomes combined with area under the ROC curve (AUC). High sensitivity permits people to be confidently regarded as not having a condition if their screening test is negative. A high negative predictive value is desirable when false negatives are minimized [32]. Differences across the ROC curves were examined using the *Z* value [33, 34]. *P* < 0.05 was considered statistically significant. Data analyses were performed using SPSS version 25.0 for Windows.

## 2.1 Ethical approval statement

The research protocol was conducted in accordance with the Helsinki declaration and was approved by Ege University Medical Research Ethical Committee (Ethics committee decision no: 19-12T/48). Verbal informed consent was obtained from all participants by a systematic and standardized process used in the geriatric medicine outpatients clinic where the study was performed. Participants or their legal guardians, when appropriate, were informed that their medical information may be used for research purposes. If they disagreed, they informed the physician taking care of them and a note was recorded in their chart. No refusal was recorded for this study.

Written consent was not taken because this research study was conducted retrospectively from data obtained for clinical purposes. Data were taken from hospital records anonymously and de-identified prior to analysis.

## 3. Results

### 3.1 General characteristics of the patients

This study enrolled 412 patients; 197 with DM, and 215 without DM. There were no significant differences for sex and age between the groups (p>0.05). The rate of low grip strength by normative regional thresholds was statistically different between DM and non-DM group (p = 0.038). Activities of Daily Living (ADL) and Instrumental Activities of Daily Living (IADL) scores were similar between groups (p>0.05). According to the median BMI of both groups, they were both consisted of mostly normal to overweight individuals with only one

participant in DM group and four participants in non-DM group identified low-weight (BMI<18.5). The characteristics of the patients according to the presence of DM are shown in **Table 1**.

### 3.2 Logistic regression analysis for disability with different hand grip strength thresholds

In multiple regression analysis, all handgrip strength thresholds were strongly associated with disability in terms of ADL and IADL in patients with Diabetes Mellitus (**Table 2**). In model 2 and model 3 with calculated regional and normative regional thresholds, polypharmacy was an important indicator for both ADLs and IADLs disability.

### 3.3 The sensitivity/negative predictive values for ADLs and IADLs disability

In DM group, the sensitivity/negative predictive value for predicting ADLs disability was similar for regional calculated and regional normative thresholds, and they were 97.7%/0.98. In non-DM group, the higher sensitivity/negative predictive value for predicting ADLs disability was with normative regional threshold, and they were 94.4% and 0.97, respectively. For IADLs; the higher sensitivity/ negative predictive value was 92.2%/0.90 by regional normative

**Table 1. Characteristics of the study population including the patients with Diabetes Mellitus (DM) and without Diabetes Mellitus (Non-DM).**

| Variables | | DM group (n = 197) | Non-DM group (n = 215) | p value |
|---|---|---|---|---|
| Age, y | | 73.5 (65–95) | 74 (65–95) | 0.788 |
| Gender, n (%) | | | | |
| | Female | 121 (61.4) | 148 (68.8) | 0.114 |
| | Male | 76 (38.6) | 67 (31.2) | |
| Hypertension, n (%) | | 136(69) | 130 (60.5) | 0.069 |
| Hyperlipidemia, n (%) | | 29 (14.7) | 18 (8.4) | 0.043[a*] |
| Cardiovascular disease, n (%) | | 82 (41.6) | 29 (13.5) | <0.001[a*] |
| Medication number | | 7 (1–17) | 4 (0–14) | <0.001[b*] |
| BMI (kg/m$^2$) | | 29.3 (16.6–56.2) | 28.2 (16.7–63.6) | 0.001[b*] |
| GS (kg) | | 21 (5–45) | 22 (5–44) | 0.806 |
| Low GS thresholds, n (%) | | | | |
| | EWGSOP2 | 81 (41.1) | 73 (34) | 0.133 |
| | Calculated regional | 147 (74.6) | 144(67) | 0.089 |
| | Normative regional | 146 (74.1) | 139 (64.7) | 0.038[a*] |
| Functional disability | | | | |
| | ADLs | 44 (22.3) | 36 (16.7) | 0.152 |
| | IADLs | 64 (32.5) | 90 (41.9) | 0.049[a*] |
| FBG (mg/dL) | | 134 (60–409) | 98 (66–125) | <0.001[b*] |
| Creatinine (mg/dL) | | 0.9 (0.5–6.5) | 0.83 (0.5–1.98) | <0.001[b*] |
| HbA1c (%) | | 6.6 (5.2–16.2) | 5.7 (4.5–6.4) | <0.001[b*] |

Characteristics of the study population including the patients with Diabetes Mellitus and without Diabetes Mellitus are explained in Table 1.

[a*] p < 0.05 obtained using Chi-Square ($x^2$) test or Fisher's exact test

[b*] p < 0.05 obtained using Mann Whitney U test

ADLs; Activities of Daily Living, IADLs; Instrumental Activities of Daily Living, EWGSOP2; Revised European Working Group on Sarcopenia in Older People, FBG; fasting blood glucose, GS; Grip strength, BMI; body mass index, HbA1c; hemoglobin A1c

**Table 2. Logistic regression analysis of disability and different hand grip strength thresholds for patients with Diabetes mellitus.**

| Variables | Disability | | | | | |
| --- | --- | --- | --- | --- | --- | --- |
| | ADLs | | | IADLs | | |
| | OR | CI (95%) | P | OR | CI (95%) | P |
| **Model 1** | | | | | | |
| Hyperlipidemia | 1.09 | 0.35–3.42 | 0.877 | 1.63 | 0.61–4.40 | 0.333 |
| Cardiovascular disease | 1.36 | 0.59–3.17 | 0.471 | 0.85 | 0.40–1.82 | 0.673 |
| Polypharmacy | 2.39 | 0.79–7.17 | 0.121 | 1.97 | 0.84–4.61 | 0.120 |
| BMI | 1.06 | 0.99–1.12 | 0.054 | 1.03 | 0.97–1.08 | 0.360 |
| EWGSOP2 consensus | 0.15 | 0.06–0.34 | **<0.001** | 0.16 | 0.08–0.33 | **<0.001** |
| **Model 2** | | | | | | |
| Hyperlipidemia | 0.97 | 0.32–2.90 | 0.956 | 1.25 | 0.50–3.16 | 0.631 |
| Cardiovascular disease | 1.51 | 0.67–3.40 | 0.326 | 0.99 | 0.49–2.01 | 0.970 |
| Polypharmacy | 2.98 | 1.01–8.73 | **0.047** | 2.41 | 1.06–5.45 | **0.035** |
| BMI | 1.05 | 0.99–1.11 | 0.096 | 1.01 | 0.96–1.06 | 0.636 |
| Calculated regional | 0.06 | 0.01–0.42 | **0.005** | 0.22 | 0.09–0.56 | **0.002** |
| **Model 3** | | | | | | |
| Hyperlipidemia | 1.06 | 0.35–3.15 | 0.92 | 1.34 | 0.53–3.41 | 0.537 |
| Cardiovascular disease | 1.32 | 0.58–2.97 | 0.51 | 0.87 | 0.42–1.78 | 0.698 |
| Polypharmacy | 3.12 | 1.06–9.16 | **0.04** | 2.50 | 1.10–5.70 | **0.029** |
| BMI | 1.07 | 1.00–1.13 | **0.04** | 1.03 | 0.98–1.08 | 0.347 |
| Normative regional | 0.05 | 0.00–0.36 | **0.003** | 0.15 | 0.06–0.43 | **<0.001** |

ADLs; Activities of Daily Living, IADLs; Instrumental Activities of Daily Living, EWGSOP2; Revised European Working Group on Sarcopenia in Older People, BMI; body mass index

thresholds in DM group; and 88.9%/0.86 by regional calculated in non-DM group. The results are shown in **Tables 3 and 4**.

### 3.4 Receiver operating characteristic (ROC) analysis results for ADLs and IADLs disability

The area under the ROC curve (AUCs) for the EWGSOP2, regional calculated, and regional normative thresholds had fair accuracy for predicting ADLs and IADLs disability both in DM and non-DM groups. Although the highest AUC values were with EWGSOP2 for ADLs and IADLs, there was no significant difference between the thresholds' AUC values. Differences across the ROC curves were examined using the Z values and p values for the patients with DM and without DM in terms of ADLs and IADLs regarding the three grip strength cut-offs. The Z and p values were shown at **the S1 Appendix**. ROC curves for ADL disability by three different HGS thresholds were shown at the **Fig 2**. ROC curves for IADL disability by three different HGS thresholds were shown at the **Fig 3**.

## 4. Discussion

Functional disability in patients with DM affects the quality of life of patients, treatment options, adherence to behavioral treatments and lifestyle changes. Muscle weakness threatens functionality in older patients with DM. Among the suggested regional thresholds for muscle weakness, this study compared the regional selected normative grip strength cut-offs [20], the regional calculated thresholds derived from walking speed [19], and the EWGSOP2 consensus normative thresholds [16] in terms of functionality in order to investigate which thresholds

**Table 3. Receiver operating characteristic analysis for the three grip strength thresholds to predict Activities of Daily Living disability.**

| | Grip Strength thresholds | | | | | |
| --- | --- | --- | --- | --- | --- | --- |
| | Non-DM group | | | DM group | | |
| | EWGSOP2 consensus | Calculated regional | Normative regional | EWGSOP2 consensus | Calculated regional | Normative regional |
| Sensitivity (%) | 72.2 | 92 | 94.4 | 75 | 97.7 | 97.7 |
| Specificity (%) | 73.7 | 38 | 41.3 | 70 | 32 | 32.7 |
| Positive predictive value (%) | 35.6 | 23 | 24.5 | 40.7 | 29.2 | 29.4 |
| Negative predictive value (%) | 93 | 96 | 97.4 | 90.5 | 98 | 98 |
| ROC AUC* (95% CI), significance (p) | 0.730 (0.637–0.822) <0.001[a] | 0.648 (0.562–0.735) 0.005[a] | 0.679 (0.597–0.761) 0.001[a] | 0.718 (0.632–0.804) <0.001[a] | 0.649 (0.568–0.730) 0.003[a] | 0.652 (0.571–0.733) 0.002[a] |

*AUC indicates area under the ROC curve. P values for the comparison of AUCs are shown in Tables 1–4 of S1 Appendix.

[a] p< 0.05 was considered statistically significant. Differences across the ROC curves were examined using the *Z* value.

ROC; Receiver operating curve, AUC; Area under curve, EWGSOP2; Revised European Working Group on Sarcopenia in Older People

best predict disability in older patients with DM. To the best of our knowledge, this is the first study in older patients with DM comparing different grip strength thresholds driven with different methodology in terms of functionality. We suggest the use of regional normative thresholds for the assessment of functional disability in older Turkish patients with DM, with normal to overweight and obese body compositions.

Patients with DM showed higher prevalence of muscle weakness than general population studies using same thresholds which might have sourced from the contribution of DM [35–39]. Regarding Turkish thresholds; for the normative regional [20], and regional calculated thresholds [19], 74.1%, and 74.6% of our study group had low grip strength values, but the prevalence was 41.1% with EWGSOP2 thresholds. Prevalence of muscle weakness was significantly higher in DM group with the normative regional thresholds than non-DM group. Recent studies have reported the prevalence of muscle weakness by EWGSOP2 ranging from 9% to 43% in community-dwelling outpatients with DM similar with our results [40, 41]. In non-DM group, muscle weakness prevalences with both regional and EWGSOP2 thresholds were; 64.7% by normative, 67% by calculated, and 34% by EWGSOP2. Fewer patients with

**Table 4. Receiver operating characteristic analysis for the three grip strength thresholds to predict Instrumental Activities of Daily Living disability.**

| | Grip Strength thresholds | | | | | |
| --- | --- | --- | --- | --- | --- | --- |
| | Non-DM group | | | DM group | | |
| | EWGSOP2 consensus | Calculated regional | Normative regional | EWGSOP2 consensus | Calculated regional | Normative regional |
| Sensitivity (%) | 61.1 | 88.9 | 87.8 | 68.8 | 90.6 | 92.2 |
| Specificity (%) | 85.6 | 48.8 | 52 | 72.2 | 33 | 34.6 |
| Positive predictive value (%) | 75.3 | 55.6 | 56.8 | 54.3 | 39.4 | 40.4 |
| Negative predictive value (%) | 75.4 | 86 | 85.5 | 82.7 | 88 | 90.2 |
| ROC AUC* (95% CI), significance (p) | 0.734 (0.663–0.804) <0.001[a] | 0.688 (0.618–0.759) <0.001[a] | 0.699 (0.629–0.769) <0.001[a] | 0.705 (0.625–0.784) <0.001[a] | 0.619 (0.539–0.698) 0.007[a] | 0.634 (0.556–0.712) 0.002[a] |

*AUC indicates area under the ROC curve. P values for the comparison of AUCs are shown in Tables 1–4 of S1 Appendix.

[a] p< 0.05 was considered statistically significant. Differences across the ROC curves were examined using the *Z* value.

ROC; Receiver operating curve, AUC; Area under curve, EWGSOP2; Revised European Working Group on Sarcopenia in Older People

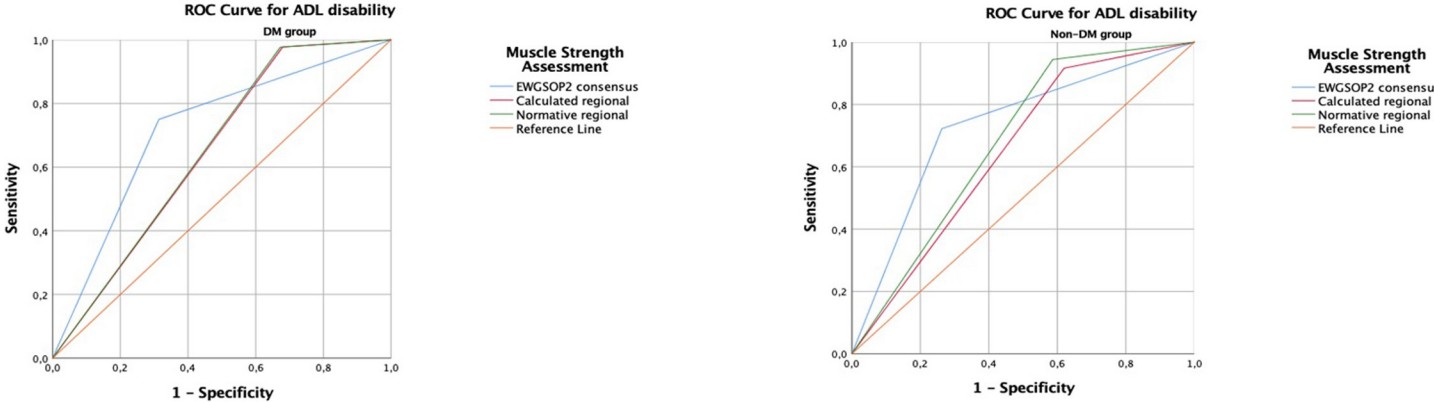

**Fig 2.** ROC Curves for Activities of Daily Living disability by EWGSOP2, calculated regional, and normative regional grip strength thresholds in the *a) DM group*, *and b) non-DM group*. ROC Curves for Activities of Daily Living disability by EWGSOP2, calculated regional, and normative regional grip strength thresholds in the patients with and without *DM* are shown in Fig 2. The AUCs for the EWGSOP2, regional calculated, and regional normative thresholds had fair accuracy for predicting ADLs disability both in DM and non-DM groups. There was no significant difference between the thresholds' AUC values. Thresholds for low grip strength: a. EWGSOP2 thresholds, b. Calculated Regional threshold, c. Normative Regional threshold. DM; diabetes mellitus, ROC; Receiver operating curve, ADLs; Activities of Daily Living, EWGSOP2; Revised European Working Group on Sarcopenia in Older People. ROC Curves are obtained by SPSS version 25.0 for Windows. Definition: Coloured graphics with no shading.

muscle weakness were diagnosed by EWGSOP2 criteria in our study in accordance with previous studies showing that regional values were with higher low grip strength rates [39, 41]. In a study conducted in older diabetic Brazilians, 21.9% of patients had low grip strength by EWGSOP2 thresholds with the Jamar dynamometer [40]. This rate was lower than general older populations of different nations by EWGSOP2 grip strength thresholds utilizing pneumatic or hydraulic dynamometers [36, 37]. Although prevalence of muscle weakness in diabetic older Brazilians by Jamar [40] was higher than previous studies in Turkey conducted by Jamar and Takei dynamometers in Turkish general older and normal to obese individuals with metabolic syndrome and endocrinological problems, it is lower than the low grip strength rate of diabetic older participants in our study by EWGSOP2 [35, 39]. Differences in prevalence across studies

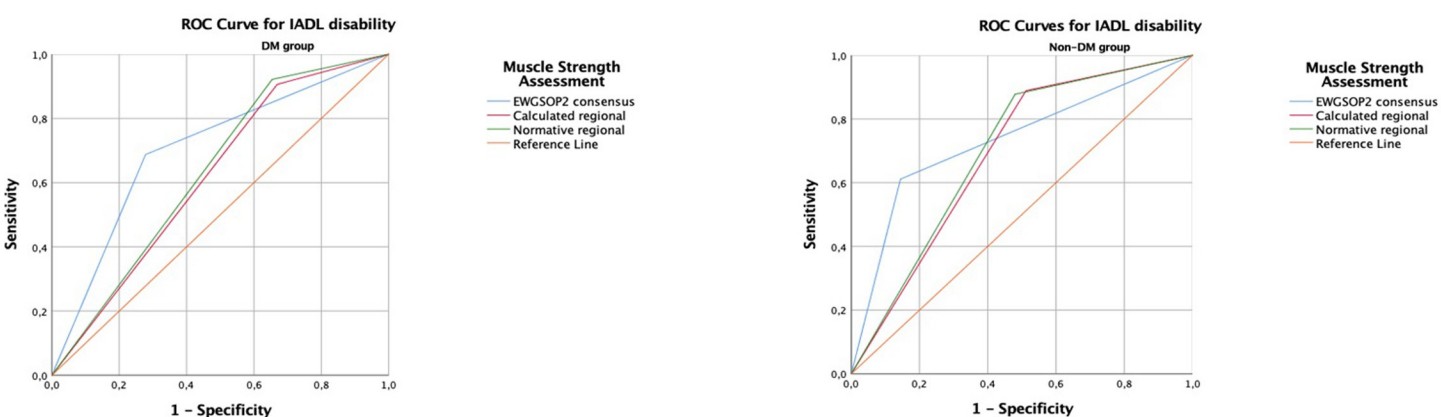

**Fig 3.** ROC Curves for Instrumental Activities of Daily Living disability by EWGSOP2, calculated regional, and normative regional grip strength thresholds in the *a) DM group*, *and b) non-DM group*. ROC Curves for Instrumental Activities of Daily Living disability by EWGSOP2, calculated regional, and normative regional grip strength thresholds in the patients with and without *DM* are shown in Fig 3. The AUCs for the EWGSOP2, regional calculated, and regional normative thresholds had fair accuracy for predicting IADLs disability both in DM and non-DM groups. There was no significant difference between the thresholds' AUC values. Thresholds for low grip strength: a. EWGSOP2 thresholds, b. Calculated Regional threshold, c. Normative Regional threshold. DM; diabetes mellitus, ROC; Receiver operating curve, IADL; Instrumental Activities of Daily Living, EWGSOP2; Revised European Working Group on Sarcopenia in Older People. ROC Curves are obtained by SPSS version 25.0 for Windows. Definition: Coloured graphics with no shading.

cannot be explained solely by overestimation of grip strength by Takei dynamometer than Jamar dynamometer [42]. Muscle weakness measurements vary significantly according to different cut-off points and dynamometers as well as the characteristics of the population, particularly comorbidities such as DM. Although there are studies on the relationship between grip strength and functionality in general population [10, 43], the research in older patients with DM is scarce [9, 13]. Besides, present studies are mostly on the relationship of grip strength with physical-performance measures or other factors such as frailty [2, 12]. In a study by Bahat et al., low grip strength values by EWGSOP2 and normative regional thresholds were related to impaired physical-performance measures and frailty in older patients [39]. In another study, patients with DM presented more deficits in ADLs and IADLs, lower grip strength values, and worse performance than patients without DM [44]. In our study, ADLs and IADLs scores, grip strength values, and ADL disability were similar in patients both with and without DM. Diabetes was associated with an increased odds of difficulties with ADLs and IADLs compared to those without diabetes [45]. McGrath et al. investigated effects of muscle weakness and diabetes on disability with ADLs [9]. The combination of muscle weakness and diabetes was associated with ADLs disability compared with those were only diabetic or weak [9]. In a study in older patients with DM, there was significant difference between dynapenic and non-dynapenic participants in terms of ADLs and IADLs [13]. In our study in older patients with DM all the three handgrip strength thresholds were strongly associated with disability in terms of ADLs and IADLs. Though ADL and IADL scores were similar between DM and non-DM groups in our study, normative regional thresholds predicted functional disability in terms of both ADLs and IADLs better than other thresholds in the group with DM. In this study, we focused on sensitivity and negative predictive value combined with AUC to assess the predictivity of HGS thresholds for functional disability. AUC can be an indicator of overall accuracy [46], and all AUC values of the thresholds' showed fair accuracy. Though the highest AUC values were with the EWGSOP2 thresholds both in DM and non-DM groups, there was not statistically significant difference between AUC values of the thresholds. While there was no statistical superiority between AUC values, the sensitivity, specificity, positive predictive value, and negative predictive value values of three grip strength cut-off values were evaluated. Sensitivity and specificity should not be relied on when making decisions in screening situations [32]. Negative predictive value and positive predictive value should also take place for making decisions for the screening tests. The approach involves the extent to which false positive and false negative results are tolerable or acceptable [32]. When we evaluate the patient to be dependent, we prescribe appropriate exercise and nutrition regimens to ensure independence. We preferred to avoid evaluating a dependent person as independent false. So, sensitivity and negative predictive value were chosen to assess the predictivity of grip strength cut-offs. Bahat et al. found regional calculated thresholds were not associated with ADLs and IADLs among community-dwelling older outpatients [38]. Regional calculated thresholds were derived from gait speed values. 6–25% of diabetic older outpatients with normal muscle strength were identified with lower physical performance [40, 41]. One reason of slow gait speed in patients with DM might be diabetic polyneuropathy that originates initially in lower limb extremities [47]. Thus, slow gait speed in patients with DM might have several causes. So, regional thresholds derived from gait speed might vary according to complications in patients with DM. Diabetes mellitus through insulin resistance, hyperglycemia, inflammation, and oxidative stress, impairs protein metabolism, vascular function, and mitochondrial health, weakening muscle integrity and strength [4, 48]. On the other hand, the decrease of muscle mass accelerates DM progression by limiting glucose disposal and worsening insulin resistance. The accumulation of fat within muscles amplifies inflammation, further fueling this cycle of metabolic dysfunction [48] This reciprocal interplay of muscle deterioration and glucose metabolism dysfunction

highlights the interconnected nature of these conditions, creating a vicious cycle that impacts both sarcopenia and DM. In a study by Bulut et al., with different normative regional thresholds calculated as two SDs below the mean of healthy young Turkish participants, sarcopenia was found to be related to ADLs impairment, balance, and frailty for both genders, and to slow walking speed for women, as well [21]. Low grip strength thresholds were determined by two different methods in our threshold choices; normative regional cut-off points derived from young reference population [20], and regional cut-off points derived from gait speed by ROC analyses [19]. So, findings of our study are consistent with these studies, and normative regional thresholds (low grip strength thresholds derived as 2.5 SDs below the gender-specific peak mean grip strength value) seem to predict disability better in older patients with DM.

## 4.1 Limitations and strengths

This is the first study investigating both ADLs and IADLs in relation to muscle weakness with different regional grip strength thresholds with different methods in comparison with EWG-SOP2 thresholds in older patients with DM. Since all participants were derived from a university hospital, there would be selection bias and limited generalizability. Another important limitation is that antidiabetic drugs that may have an effect on glycemic control and functionality were not evaluated in this study. Frailty is also one of the most important predisability conditions that is prevalent in diabetic older adults. Due to the retrospective design of the study, we were not able to assess patients for frailty, as the available patient records did not contain complete data on frailty parameters.

In older patients with DM, muscle weakness and functional disability are related topics that may affect lifestyle change and treatment management, and muscle weakness might be an important indicator for functional disability. For this reason, it is important to point out an objective grip strength threshold for functional disability. Multi-center and multi-national studies that take into account the points mentioned in the limitation part may lead us to reach more generalizable results. So, the results of our study need to be investigated in different ethnicities, and samples.

## 5. Conclusion

Functionality is substantial for geriatric individuals. Evaluating functional disability is particularly essential in older patients with DM as it takes place in most of the decision-making processes for the treatment and follow-up such as the choice of therapy for management of DM, monitoring medications and glycemic control as well as adherence to walking and exercise recommendations. In this context, grip strength thresholds that best predict functional disability in older patients with DM is substantial. In terms of the criteria for the predictivity of each threshold, regional thresholds showed higher accuracy for functional dependency in both DM and non-DM groups. Normative regional thresholds predicted disability in older patients with DM in terms of both ADLs and IADLs. We suggest using regional normative thresholds for the assessment of functional disability in older Turkish patients with DM, with normal to overweight and obese body compositions. Further longitudinal studies using different suggested regional grip strength thresholds derived by diverse methods in comparison with EWGSOP2 thresholds in diabetic patients with different comorbidities, body compositions, and ethnicities are needed.

## Supporting information

**S1 Appendix. p and z values of the comparisons between the AUCs for grip strength thresholds for ADLs and IADLs in the DM and non-DM groups.**
(DOCX)

**S1 Data. The hand grip strength values of the first 100 samples of the DM group as a subset of the anonymised data.**
(SAV)

## Author Contributions

**Conceptualization:** Sibel Çavdar, Fatma Ozge Kayhan Kocak, Sumru Savas.

**Data curation:** Sibel Çavdar, Fatma Ozge Kayhan Kocak.

**Formal analysis:** Sibel Çavdar, Fatma Ozge Kayhan Kocak, Sumru Savas.

**Investigation:** Sibel Çavdar, Fatma Ozge Kayhan Kocak, Sumru Savas.

**Methodology:** Sibel Çavdar, Fatma Ozge Kayhan Kocak, Sumru Savas.

**Project administration:** Sibel Çavdar, Fatma Ozge Kayhan Kocak, Sumru Savas.

**Supervision:** Sibel Çavdar, Fatma Ozge Kayhan Kocak, Sumru Savas.

**Validation:** Sibel Çavdar, Fatma Ozge Kayhan Kocak, Sumru Savas.

**Visualization:** Sibel Çavdar, Fatma Ozge Kayhan Kocak, Sumru Savas.

**Writing – original draft:** Sibel Çavdar.

**Writing – review & editing:** Sibel Çavdar, Fatma Ozge Kayhan Kocak, Sumru Savas.

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
