## [Decision Letter · Decision Letter 0]

26 Nov 2024

PONE-D-24-45322Regional normative thresholds predict functional disability in older Turkish patients with Diabetes Mellitus better than EWGSOP2 and regional calculated thresholdsPLOS ONE

Dear Dr. Çavdar,

Thank you for submitting your manuscript to PLOS ONE. After careful consideration, we feel that it has merit but does not fully meet PLOS ONE’s publication criteria as it currently stands. Therefore, we invite you to submit a revised version of the manuscript that addresses the points raised during the review process.

We look forward to receiving your revised manuscript.

Kind regards,

Masoud Rahmati

Academic Editor

PLOS ONE

Journal Requirements:

1. When submitting your revision, we need you to address these additional requirements. Please ensure that your manuscript meets PLOS ONE's style requirements, including those for file naming. The PLOS ONE style templates can be found at https://journals.plos.org/plosone/s/file?id=wjVg/PLOSOne_formatting_sample_main_body.pdf and 
https://journals.plos.org/plosone/s/file?id=ba62/PLOSOne_formatting_sample_title_authors_affiliations.pdf

2. We note that you have indicated that there are restrictions to data sharing for this study. PLOS only allows data to be available upon request if there are legal or ethical restrictions on sharing data publicly. For more information on unacceptable data access restrictions, please see http://journals.plos.org/plosone/s/data-availability#loc-unacceptable-data-access-restrictions. Before we proceed with your manuscript, please address the following prompts: a) If there are ethical or legal restrictions on sharing a de-identified data set, please explain them in detail (e.g., data contain potentially identifying or sensitive patient information, data are owned by a third-party organization, etc.) and who has imposed them (e.g., a Research Ethics Committee or Institutional Review Board, etc.). Please also provide contact information for a data access committee, ethics committee, or other institutional body to which data requests may be sent. b) If there are no restrictions, please upload the minimal anonymized data set necessary to replicate your study findings to a stable, public repository and provide us with the relevant URLs, DOIs, or accession numbers. For a list of recommended repositories, please see https://journals.plos.org/plosone/s/recommended-repositories. You also have the option of uploading the data as Supporting Information files, but we would recommend depositing data directly to a data repository if possible. We will update your Data Availability statement on your behalf to reflect the information you provide.

Reviewers' comments:

Reviewer's Responses to Questions

**Comments to the Author**

1. Is the manuscript technically sound, and do the data support the conclusions?

Reviewer #1: Yes

Reviewer #2: Yes

2. Has the statistical analysis been performed appropriately and rigorously? 

Reviewer #1: No

Reviewer #2: I Don't Know

3. Have the authors made all data underlying the findings in their manuscript fully available?

Reviewer #1: No

Reviewer #2: Yes

4. Is the manuscript presented in an intelligible fashion and written in standard English?

Reviewer #1: Yes

Reviewer #2: Yes

5. Review Comments to the Author

Reviewer #1: I would like to commend the authors for their valuable and impactful research. Your work addresses an important topic and holds significant potential for application in clinical settings. DM is a prevalent condition that is related to disability, the early detection of muscle weakness can be useful as a preventive goal for disability.

However, there are few concerns that need to be addressed

1. Data availability statement:

If possible, consider sharing a subset of anonymized data to help others replicate your study while respecting the constraints.

2. Regarding exclusion criteria:

Apart from neuropathy, did the authors exclude conditions that cause hand pain, or deformity?

3. The EWGSOP2 consensus thresholds for HGS is based on the measurement by the Jamar dynamometer.

The Takei T.K.K. 5401 digital dynamometer underwent validation against Jamar hydraulic dynamometer and although there was good intraclass correlation coefficient but the minimal detectable change was 10.14%. According to the Bland-Altman plot, the Takei dynamometer overestimated GS measurements by 0.85 kg (bias: −0.85, 95% CI −1.47 to −2.22). Low GS prevalences according to the EWGSOP2 thresholds [1] by the Jamar, and Takei dynamometers were 6.4%, and 4.5% in all participants, 4.8%, 0% in women, 8.3%, 10.4% in men, respectively. And the authors concluded that Takei dynamometer overestimated GS over the Jamar dynamometer.

Savas, S.; Kilavuz, A.; Kayhan Koçak, F.Ö.; Cavdar, S. Comparison of Grip Strength Measurements by Widely Used Three Dynamometers in Outpatients Aged 60 Years and Over. J. Clin. Med. 2023, 12, 4260. https://doi.org/10.3390/jcm12134260

4. Were the sample participants were assessed for frailty phenotype? Frailty is one of the most important predisability conditions that is prevalent in diabetic older adults that include muscle weakness as a core domain for diagnosis.

5. Discussion needs to be supplemented with pathophysiological explanation for the presented associations.

6. Regression analysis is needed to adjust for HL , CVD, poly pharmacy, and BMI as possible predictor for disability and muscle weakness

7. The figures need to be of higher quality

Reviewer #2: Regional normative thresholds predict functional disability in older Turkish patients with Diabetes Mellitus better than EWGSOP2 and regional calculated thresholds

This study examined the predictivity of muscle weakness defined as low grip strength thresholds determined by EWGSOP2, and two regional thresholds in older patients with Diabetes Mellitus for functional disability. Conclusion was that regional normative thresholds predicted both ADL and IADL disability in older patients with Diabetes Mellitus. The topic of this study is interesting. However, I do have some points in the text that I would like to raise.

Authors have used a lot of abbreviations throughout the manuscript and in tables as well in figures. Excessive use of abbreviations makes the text confusing and reading is difficult. As it is mentioned in the submission guidelines of the journal the number of abbreviations should keep to a minimum. I suggest that Authors think carefully what abbreviation are necessary in this manuscript. For example, grip strength does not need abbreviation and in the Table 1 diseases such as HT, HL, CVD can also be written as a whole.

Similarly, the title of the manuscript uses the abbreviation EWGSOP2. Authors cannot assume that all readers know what the abbreviation means. Could this EWGSOP2 be written at least a little open in the title or then modified the title again without the acronym.

Line 75, Introduction. Authors mentioned the purpose of this study. I suggest mentioning already here that three different thresholds will be used and compared in this study.

Authors use the EWGSOP2 one of the thresholds. I am not sure if all readers know what is this EWGSOP2. When this is mentioned at the first time in the line 66, I strongly suggest telling briefly what this is and first write it as a whole and the give the abbreviation.

Page 4, line 96, Materials and methods. Socio demographic data were also noted in all patients. What socio demographic data were used in this study, this information could write here.

Page 4, line 100. Authors mentioned that grip strength measurement protocol was defined in three references (22-24). However, could you briefly write also here the details of the protocol. Is it maximum strength out of three trial or what?

Page 4, line 96. Authors mention abbreviation BMI here at the first time. The abbreviation should be written and BMI in the brackets.

Unfortunately, I am not the expert of these kind statistical methods, so I have to leave comments about statistical methods on another reviewers. However, Authors have explained well the levels of specificity, sensitivity, AUC in the text.

Table 1 shows that there were statistically significant differences between groups (DM and Non-DM) in diseases, number of medications, BMI, hemoglobin and fasting blood glucose and creatine. However, Authors did not mention this. Is it possible that people in the group of DM could have more diseases that have effect on results?

Also, on tables Authors use a lot of abbreviation, too much. I would modify little the topic of the Table 1: Characteristics of the study population including the patients with Diabetes Mellitus (DM) and 147 without Diabetes Mellitus (Non-DM).

The appearance of Table 1 and Tables 2 and 3 is different.

The quality of figures in its current form is poor and makes it difficult to read. Will they be corrected if the manuscript is accepted for publication?

Page 13, line 298. The Authors conclude: We suggest using regional 299 normative thresholds for the assessment of functional disability in older Turkish patients with DM, with 300 normal to overweight and obese body compositions. But what about internationally. What does this study with Turkish patients gives internationally?

6. PLOS authors have the option to publish the peer review history of their article (what does this mean?). If published, this will include your full peer review and any attached files.

Reviewer #1: **Yes: **Doha Rasheedy

Reviewer #2: No

---

## [Author Response · Author response to Decision Letter 0]

7 Dec 2024

Journal Requirements:

1. When submitting your revision, we need you to address these additional requirements. Please ensure that your manuscript meets PLOS ONE's style requirements, including those for file naming. The PLOS ONE style templates can be found at https://journals.plos.org/plosone/s/file?id=wjVg/PLOSOne_formatting_sample_main_body.pdf and https://journals.plos.org/plosone/s/file?id=ba62/PLOSOne_formatting_sample_title_authors_affiliations.pdf

A1: It is formatted as requested.

2. We note that you have indicated that there are restrictions to data sharing for this study. PLOS only allows data to be available upon request if there are legal or ethical restrictions on sharing data publicly. For more information on unacceptable data access restrictions, please see http://journals.plos.org/plosone/s/data-availability#loc-unacceptable-data-access-restrictions. Before we proceed with your manuscript, please address the following prompts: a) If there are ethical or legal restrictions on sharing a de-identified data set, please explain them in detail (e.g., data contain potentially identifying or sensitive patient information, data are owned by a third-party organization, etc.) and who has imposed them (e.g., a Research Ethics Committee or Institutional Review Board, etc.). Please also provide contact information for a data access committee, ethics committee, or other institutional body to which data requests may be sent. b) If there are no restrictions, please upload the minimal anonymized data set necessary to replicate your study findings to a stable, public repository and provide us with the relevant URLs, DOIs, or accession numbers. For a list of recommended repositories, please see https://journals.plos.org/plosone/s/recommended-repositories. You also have the option of uploading the data as Supporting Information files, but we would recommend depositing data directly to a data repository if possible. We will update your Data Availability statement on your behalf to reflect the information you provide.

A2: We checked the data availability statements for the journal and our data availability statement is in line with the journal’s Acceptable Data Access Restrictions. Data is available from the secretary of the Ege University Ethics Committee via email (egetaek@gmail.com) for researchers who meet the criteria for access to confidential data. 

A3: Full ethics statement is placed in the ‘Methods’ section. In this section, all requested information is given.

Reviewers' Comments to the Authors and Authors’ Answers

Thank you very much for giving us the opportunity to revise our manuscript. I would like to thank the editor and the reviewers for their valuable comments. Insertions or corrections is highlighted in the manuscript.

Reviewer #1: 

I would like to commend the authors for their valuable and impactful research. Your work addresses an important topic and holds significant potential for application in clinical settings. DM is a prevalent condition that is related to disability, the early detection of muscle weakness can be useful as a preventive goal for disability.

However, there are few concerns that need to be addressed

Q1. Data availability statement:If possible, consider sharing a subset of anonymized data to help others replicate your study while respecting the constraints.

A1. In accordance with the reviewer's comments, we have shared the hand grip strength values of the first 100 samples of the DM group as a subset of the anonymised data. There are ethical limitations to publicly sharing a de-identified dataset due to sensitive patient information. Our data availability statement is in line with the journal's acceptable data access restrictions. The data are available from the secretary of the Ege University Ethics Committee by e-mail (egetaek@gmail.com) for researchers who meet the criteria for access to confidential data.

Q2. Regarding exclusion criteria:Apart from neuropathy, did the authors exclude conditions that cause hand pain, or deformity?

A2. Thank you for giving us the opportunity to clarify the exclusion criteria. Since we do not measure handgrip strength in our practice for patients with severe pain and deformity in the hands, we neglected to mention about this in the exclusion criteria. ‘’Patients with severe pain, arthritis, or deformity in the hand joints, which could compromise handgrip strength, were excluded from the study as handgrip strength measurements were not conducted in such cases.’’ This sentence is added to the materials and methods section.

Q3. The EWGSOP2 consensus thresholds for HGS is based on the measurement by the Jamar dynamometer.

The Takei T.K.K. 5401 digital dynamometer underwent validation against Jamar hydraulic dynamometer and although there was good intraclass correlation coefficient but the minimal detectable change was 10.14%. According to the Bland-Altman plot, the Takei dynamometer overestimated GS measurements by 0.85 kg (bias: −0.85, 95% CI −1.47 to −2.22). Low GS prevalences according to the EWGSOP2 thresholds [1] by the Jamar, and Takei dynamometers were 6.4%, and 4.5% in all participants, 4.8%, 0% in women, 8.3%, 10.4% in men, respectively. And the authors concluded that Takei dynamometer overestimated GS over the Jamar dynamometer.

Savas, S.; Kilavuz, A.; Kayhan Koçak, F.Ö.; Cavdar, S. Comparison of Grip Strength Measurements by Widely Used Three Dynamometers in Outpatients Aged 60 Years and Over. J. Clin. Med. 2023, 12, 4260. https://doi.org/10.3390/jcm12134260

A3.Thank you for pointing out our team’s study. ‘Differences in prevalence across studies cannot be explained solely by overestimation of grip strength by Takei dynamometer than Jamar dynamometer [42].’ This sentence has been added to the discussion. 

Q4. Were the sample participants were assessed for frailty phenotype? Frailty is one of the most important predisability conditions that is prevalent in diabetic older adults that include muscle weakness as a core domain for diagnosis.

A4. Thank you for highlighting this important issue. Due to the retrospective design of the study, we were unable to assess patients for frailty, as the available patient records lacked complete data on frailty parameters. We have acknowledged this as a limitation of our study based on your suggestion:

Frailty is also one of the most important predisability conditions that is prevalent in diabetic older adults. Due to the retrospective design of the study, we were not able to assess patients for frailty, as the available patient records did not contain complete data on frailty parameters.

Q5. Discussion needs to be supplemented with pathophysiological explanation for the presented associations.

A5. We would like to thank the reviewer for vigilant reading. In accordance with the reviewer’s comments, the following paragraph has been added to the Discussion section. “Diabetes mellitus through insulin resistance, hyperglycemia, inflammation, and oxidative stress, impairs protein metabolism, vascular function, and mitochondrial health, weakening muscle integrity and strength [4, 48]. On the other hand, the decrease of muscle mass accelerates DM progression by limiting glucose disposal and worsening insulin resistance. The accumulation of fat within muscles amplifies inflammation, further fueling this cycle of metabolic dysfunction [48] This reciprocal interplay of muscle deterioration and glucose metabolism dysfunction highlights the interconnected nature of these conditions, creating a vicious cycle that impacts both sarcopenia and DM.”

Q6. Regression analysis is needed to adjust for HL , CVD, poly pharmacy, and BMI as possible predictor for disability and muscle weakness

A6. Following the reviewer's valuable comment, we performed the regression analysis adjusted for hyperlipidemia, cardiovascular disease, polypharmacy and BMI. The following paragraph about regression analysis has been added to the material method section: Multivariate analysis included parameters that were significant in presence of disability such as HL, CVD, polypharmacy and BMI. Multiple linear regression, the “enter” method, was used to define descriptive variables on disability, and to calculate the adjusted odds ratios (ORs) and 95% confidence interval (CI). 

In the results section the following sentence and the table are added to indicate the results of regression analyses: 

Logistic regression analysis for disability with different hand grip strength thresholds

In multiple regression analysis, all handgrip strength thresholds were strongly associated with disability in terms of ADL and IADL in patients with Diabetes Mellitus (table 2). In model 2 and model 3 with calculated regional and normative regional thresholds, polypharmacy was an important indicator for both ADLs and IADLs disability. Table 2 is added in the manuscript.In our study in older patients with DM all the three handgrip strength thresholds were strongly associated with disability in terms of ADLs and IADLs. This sentence is added to the discussion.

Q7. The figures need to be of higher quality

A7. We would like to thank the reviewer for valuable comments. The quality of the figures has been improved.

Figure 2 ROC Curves for Activities of Daily Living disability by EWGSOP2, calculated regional, and normative regional grip strength thresholds in the a) DM group, and b) non-DM group

Figure 3 ROC Curves for Instrumental Activities of Daily Living disability by EWGSOP2, calculated regional, and normative regional grip strength thresholds in the a) DM group, and b) non-DM group

Reviewer #2: 

This study examined the predictivity of muscle weakness defined as low grip strength thresholds determined by EWGSOP2, and two regional thresholds in older patients with Diabetes Mellitus for functional disability. Conclusion was that regional normative thresholds predicted both ADL and IADL disability in older patients with Diabetes Mellitus. The topic of this study is interesting. However, I do have some points in the text that I would like to raise.

Q1. Authors have used a lot of abbreviations throughout the manuscript and in tables as well in figures. Excessive use of abbreviations makes the text confusing and reading is difficult. As it is mentioned in the submission guidelines of the journal the number of abbreviations should keep to a minimum. I suggest that Authors think carefully what abbreviation are necessary in this manuscript. For example, grip strength does not need abbreviation and in the Table 1 diseases such as HT, HL, CVD can also be written as a whole.

A1. Thank you very much for your fair warning about abbreviations. In Table 1, abbreviations for disease names have been removed and their full forms were indicated in the table.

In the manuscript, abbreviations of ‘GS’ are replaced with the long form ‘grip strength’ as suggested. In addition we have removed the NPV and PPV which are used as abbreviations for negative predictive value and positive predictive value, and replaced them with their full forms. Thank you for your suggestions on abbreviations to make the article easier to read.

Q2. Similarly, the title of the manuscript uses the abbreviation EWGSOP2. Authors cannot assume that all readers know what the abbreviation means. Could this EWGSOP2 be written at least a little open in the title or then modified the title again without the acronym.

A2. In accordance with the reviewer’s comments, we have changed the title to “The Association of Muscle Weakness, Measured by Three Different Grip Strength Thresholds, with Functional Disability in Older Patients with Diabetes Mellitus

Q3. Line 75, Introduction. Authors mentioned the purpose of this study. I suggest mentioning already here that three different thresholds will be used and compared in this study.

A3. In accordance with the reviewer’s comments, the sentence has been revised as suggested:

“Therefore, we aimed to investigate the predictivity of muscle weakness for functional disability by three different low GS thresholds in older patients with DM in comparison with each other and also in comparison with older patients without DM. “

Q4. Authors use the EWGSOP2 one of the thresholds. I am not sure if all readers know what is this EWGSOP2. When this is mentioned at the first time in the line 66, I strongly suggest telling briefly what this is and first write it as a whole and the give the abbreviation.

A4. In line 66, the full name for EWGSOP and also a short explanation are added:

The European Working Group on Sarcopenia in Older People (EWGSOP) indicated a sarcopenia definition in 2010 and than in 2018 the group updated the definition.

Q5.Page 4, line 96, Materials and methods. Socio demographic data were also noted in all patients. What socio demographic data were used in this study, this information could write here.

A5.Added in parentheses: Socio-demographic data (age and gender)

Q6. Page 4, line 100. Authors mentioned that grip strength measurement protocol was defined in three references (22-24). However, could you briefly write also here the details of the protocol. Is it maximum strength out of three trial or what?

A6. Thank you for giving us the opportunity to clarify method. One of the 3 references was the validated protocol by the National Health and Nutrition Examination Survey. The other two references were studies using this method. However, in order to avoid confusion by giving 3 references, only the source where the original protocol was given was left and the other 2 references were deleted from that sentence. These 2 references were also cited elsewhere in the article, they still exist in the reference list. Only the order of references has changed. ‘’The maximum value of three grip strength measurements was selected.’’has been added in the methods section.

Q7.Page 4, line 96. Authors mention abbreviation BMI here at the first time. The abbreviation should be written and BMI in the brackets.

A7. We would like to thank the reviewer for vigilant reading.. It is corrected. body mass index (BMI) is added where mentioned. We had given the explanation in the results section in revised line 152, and the long form is deleted.

Q8. Unfortunately, I am not the expert of these kind statistical methods, so I have to leave comments about statistical methods on another reviewers. However, Authors have explained well the levels of specificity, sensitivity, AUC in the text.

A8. Thank you for your comment. As you mentioned, we made an effort to explain these concepts within the manuscript to ensure they are easier for everyone to understand.

Q9. Table 1 shows that there were statistically significant differences between groups (DM and Non-DM) in diseases, number of medications, BMI, hemoglobin and fasting blood glucose and creatine. However, Authors did not mention this. Is it possible that people in the group of DM could have more diseases that have effect on results?

A9. Thank you for this valuable comment. We add logistic regression analyses to clarify this question. Fasting bloog glucose, HBA1C, and Creatinine levels were not included in the analysis because they were high as a result of DM disease when compared to patients without DM.

In the results section the following sentence and a new table are added to indicate the results of regression analyses: 

Logistic regression analysis for disability with different hand grip strength thresholds

In multiple regression analysis, all handgrip strength thresholds were strongly associated with disability in terms of ADL and IADL in patients with Diabetes Mellitus (table 2) In model 2 and model 3 with calculated regional and normative regional thresholds, polypharmacy was an important indicator for both ADLs and IADLs disability.

Table is added in the manuscript.

Q10. Also, on tables Authors use a lot of abbreviation, too much. I would modify l

---

## [Decision Letter · Decision Letter 1]

26 Dec 2024

The association of muscle weakness, measured by three different grip strength thresholds, with functional disability in older patients with Diabetes Mellitus

PONE-D-24-45322R1

Dear Dr. Çavdar,

We’re pleased to inform you that your manuscript has been judged scientifically suitable for publication and will be formally accepted for publication once it meets all outstanding technical requirements.

Kind regards,

Masoud Rahmati

Academic Editor

PLOS ONE

Additional Editor Comments (optional):

Reviewers' comments:

Reviewer's Responses to Questions

**Comments to the Author**

1. If the authors have adequately addressed your comments raised in a previous round of review and you feel that this manuscript is now acceptable for publication, you may indicate that here to bypass the “Comments to the Author” section, enter your conflict of interest statement in the “Confidential to Editor” section, and submit your "Accept" recommendation.

Reviewer #1: All comments have been addressed

Reviewer #2: (No Response)

2. Is the manuscript technically sound, and do the data support the conclusions?

Reviewer #1: Yes

Reviewer #2: Yes

3. Has the statistical analysis been performed appropriately and rigorously? 

Reviewer #1: Yes

Reviewer #2: Yes

4. Have the authors made all data underlying the findings in their manuscript fully available?

Reviewer #1: No

Reviewer #2: Yes

5. Is the manuscript presented in an intelligible fashion and written in standard English?

Reviewer #1: Yes

Reviewer #2: Yes

6. Review Comments to the Author

Reviewer #1: the authors responded adequately to the reviewers' comments. changes in manuscript and figures were performed. logistic regression model of disability was performed

Reviewer #2: Thank you to the Authors, they have responded clearly and comprehensively to the questions posed.

Authors have changed the title of the manuscript, but I wonder if there are two different options for the title.

The highlighted one: The association of muscle weakness, measured by three different grip strength thresholds, with functional disability in older patients with Diabetes Mellitus

But there is also another:

The association of muscle weakness with functionality in older patients with Diabetes Mellitus

The highlighted is a little difficult to read and my suggestion is:

The association of muscle weakness with functional disability in older patients with Diabetes Mellitus: Measured by three different grip strength thresholds

7. PLOS authors have the option to publish the peer review history of their article (what does this mean?). If published, this will include your full peer review and any attached files.

Reviewer #1: No

Reviewer #2: No

---

## [Editor Report · Acceptance letter]

21 Jan 2025

PONE-D-24-45322R1 

PLOS ONE

Dear Dr. Çavdar, 

I'm pleased to inform you that your manuscript has been deemed suitable for publication in PLOS ONE. Congratulations! Your manuscript is now being handed over to our production team.

Kind regards, 

on behalf of

Dr. Masoud Rahmati 

Academic Editor

PLOS ONE